# Analysis of the Anti-Skid Properties of New Airfield Pavements in Aspect of Applicable Requirements

**Mariusz Zieja** [1] , **Mariusz Wesołowski** [2] , **Krzysztof Blacha** [2,*] **and Paweł Iwanowski** [2]

1   IT Logistics Support Division, Air Force Institute of Technology, ul. Ks. Bolesława 6, 01-494 Warsaw, Poland;
    mariusz.zieja@itwl.pl
2   Airfield Division, Air Force Institute of Technology, ul. Ks. Bolesława 6, 01-494 Warsaw, Poland;
    mariusz.wesolowski@itwl.pl (M.W.); pawel.iwanowski@itwl.pl (P.I.)
*   Correspondence: krzysztof.blacha@itwl.pl; Tel.: +48-261-851-424

**Abstract:** The assessment of the anti-skid properties of airfield pavements is one of the elements in the process of determining their current technical condition, which is important in terms of the safety of air operations. The condition of the pavement should be qualified on the basis of the specified and required values (criteria), in this case, the coefficient of friction and the depth of the texture. Unfortunately, in practice, the assessment of the texture of new airfield pavements raises some doubts with regard to the existing requirements. The work presents an analysis of the results of texture tests for new airfield pavements in relation to current requirements. In addition, the authors proposed a new criterion of average texture depth for new airfield pavements, based on actual field measurement results. Field studies used an innovative method of assessing anti-skid properties with a measuring system that allows simultaneous measurement of the coefficient of friction, $\mu$, and the new continuous coefficient of average profile depth and texture *CMPTD*, from which *MPD* (Mean Profile Depth) and *ETD* (Estimated Texture Depth) can be determined. The results of the tests cast doubt on the possibility of obtaining the average texture depth (currently 1.00 mm) required for new airfield pavements.

**Keywords:** airfield pavement texture; coefficient of friction; anti-skid properties of the airfield pavement; safety of air operations; skid resistance

## 1. Introduction

The technical condition of airfield pavements is important in terms of the safety of air operations. The assessment of the anti-skid properties of airfield pavements is one of the elements in the process of assessing their current technical condition. These characteristics determine, inter alia, the adhesion of the aircraft tyre to the surface, i.e., the ability to generate friction force between the surface of the airport functional element (AFE) and the wheels of the aircraft under conditions of reciprocal skid. Until recently, the technical condition of airfield pavements in the field of anti-skid properties was based only on the coefficient of friction, measured during the roughness test. Currently, the assessment of airfield pavement's anti-skid properties also includes depth measurements of the pavement's texture. In practice, the aim is to ensure that airfield pavements have good anti-skid properties, which should be defined by specific and required values (criteria), which are the coefficient of friction and the depth of the texture in this case.

All over the world, roughness tests are carried out on the basis of the regulations of international aviation organizations, such as the European Aviation Safety Authority (EASA) [1], International Civil Aviation Organization (ICAO) [2–4] and the Federal Aviation Administration (FAA) [5], or national documents based on them. Such a national document which was developed in Poland based on many years of experience is the defence standard [6]. According to these documents, the roughness test of the AFE pavement shall be carried out with devices enabling the continuous measurement of the coefficient of friction

between the wheel of the moving aircraft and the airfield pavement surface. Devices with a smooth-retreaded tyre and a water-spraying system shall be used to measure the friction coefficient of the airfield pavement with a layer of water not less than 1 mm thick at two measuring speeds, i.e., 65 or 95 km/h. In Poland, the most popular devices for measuring the coefficient of friction of the AFE pavement are ASFT (airport surface friction tester) devices mounted on the vehicle or on a trailer. The smaller airport functional element, the geometry of which does not allow to perform measurement at 65 or 95 km/h, can be tested using the handheld portable T2Go roughness tester. The assessment of the friction characteristics of airfield pavements (with an indication of runways) shall be carried out in two aspects, i.e.,:

- Assessments of the friction characteristics of artificial pavements covered with a layer of snow, slush, ice or frost;
- Assessment of the friction characteristics of the pavements for construction and operation purposes.

When assessing the pavement according to the first aspect, the expected friction of the pavement should be classified as good, medium to good, medium, medium to bad, bad. The predicted pavement friction, based on the measured friction coefficient, with the runway covered with packed snow or ice only, may be reported in accordance with Table 1 [1].

**Table 1.** Summary of the friction coefficient values and the assessment of braking efficiency for snow-covered and ice-covered airfield pavements.

| Measured Coefficient ($\mu$) | Estimated Surface Friction | Code |
|---|---|---|
| 0.40 and above | Good | 5 |
| 0.39 to 0.36 | Medium to good | 4 |
| 0.35 to 0.30 | Medium | 3 |
| 0.29 to 0.26 | Medium to poor | 2 |
| 0.25 and below | Poor | 1 |

The frictional characteristics when assessing the pavement according to the second aspect (mainly on the runway) should be:

- Assessed to verify the friction characteristics of the pavement on a new or renovated runway;
- Periodically assessed in order to determine the slipperiness of an artificial runway.

During measurements with ASFT devices, the values of the coefficient of friction for "new pavements", "surfaces in service, beyond which corrective action is to be taken" and "minimum limit values" should be in accordance with Table 2 [6].

**Table 2.** Summary of measuring devices (ASFT) used to test the roughness of airfield pavements and the mean values of the required coefficients of friction.

| Measuring Device | Measuring Speed (km/h) | Coefficient of Friction $\mu$ | | |
|---|---|---|---|---|
| | | Design Values for New Pavements | Values for Planning Corrective Actions | Minimum Values |
| Surface Friction Tester Trailer and Airport | 65 | 0.70 | 0.50 | 0.40 |
| Surface Friction Tester Vehicle (ASFT) | 95 | 0.60 | 0.40 | 0.32 |

In addition, EASA introduced a runway surface condition assessment for maintenance purposes with the document [7] this year (2021). In this case, when using friction measuring devices, maintenance planning and minimum friction levels should be in accordance with Table 3 below. In addition, the document presents a new approach to pavement maintenance as well as maintenance planning and minimum standards. The method of monitoring the physical parameters of the runway surface is also described.

**Table 3.** Friction standards in order to assess the condition of a runway's pavement for maintenance purposes.

| Measurement Device | 65 km/h | | 95 km/h | |
|---|---|---|---|---|
| | Minimum | Maintenance Planning | Minimum | Maintenance Planning |
| Airport Surface Friction Tester | 0.50 | 0.60 | 0.34 | 0.47 |
| Dynatest Consulting Inc. Dynatest Runway Friction Tester | 0.50 | 0.60 | 0.41 | 0.54 |
| Findlay, Irvine, Ltd. Griptester Friction Meter | 0.43 | 0.53 | 0.24 | 0.36 |
| Halliday Technologies RT3 | 0.45 | 0.55 | 0.42 | 0.52 |
| Moventor Oy Inc. BV-11 Skiddometer | 0.50 | 0.60 | 0.34 | 0.47 |
| Mu Meter | 0.42 | 0.52 | 0.26 | 0.38 |
| NAC Dynamic Friction Tester | 0.42 | 0.52 | 0.28 | 0.38 |
| Norsemeter RUNAR (operated at fixed 16% slip) | 0.45 | 0.52 | 0.32 | 0.42 |
| Automatic Friction Measuring Device (Instrument de Mesure Automatique de Glissance)—IMAG | 0.30 | 0.40 | 0.20 | 0.30 |

The examination of the airport pavement's texture is currently carried out with the use of point measurement methods, i.e., the volumetric or the by-volume method (measurement of the mean texture depth *MTD*) according to PN-EN 13036-1: 2010 Road and airfield surface characteristics. Test methods. Part 1: Measurement of pavement surface macrotexture depth using a volumetric patch technique [8] or the profilometric method (measurement of the mean profile depth of the *MPD/dMPD*) according to PN-EN ISO 13473-1: 2019 Characterization of pavement texture by use of surface profiles. Part 1: Determination of mean profile depth [9]. The *MPD* value can be converted into an estimated texture depth *ETD*, for this purpose the transformation equation is used:

$$ETD = 0.2 + 0.8 \ MPD \tag{1}$$

The texture depth requirements that must be met by new airport pavements are included in the documents issued by global aviation organizations, such as EASA, ICAO and FAA, which are presented in Table 4.

**Table 4.** Texture depth requirements for new airport pavements.

| Document | Texture Depth (mm) |
|---|---|
| Annex 14 to the Convention on the International Civil Aviation, Aerodromes Volume I—Aerodrome Design and Operation (ICAO) [2] Doc. 9157 AN/901 Aerodrome Design Manual Part 1—Runways (ICAO) [3] Easy Access Rules for Aerodromes (Regulation (EU) No 139/2014) (EASA) [1] | $\geq$1.00 |
| Doc. 9137 AN/898 Airport Service Manual Part 2—Pavement Surface Conditions (ICAO) [4] | <1.00 mm |
| Advisory Circular no: 150/5320-12C, U.S. Department of Transportation (FAA) [5] | $\geq$1.14 mm |

In terms of texture depth, the applicable aviation documents define the requirements practically only for new airport pavements and they mainly relate to the runway surface; what is more, they are not unambiguous, as is presented in Table 3. When reviewing them, the question arises: what minimum required value should new pavements adopt—1.0 or 1.14 mm, but also how to treat the provision allowing a value below 1.0 mm. On the other hand, for the assessment of older airport pavements, only the ESDU (Engineering Sciences Data Unit) classification developed for the runway based on the information on the texture presented in Table 5 [1] can be used at the moment. It is assumed that the 0.25 mm texture depth should ensure safe flight operations. It should be noted that the texture depth requirements for airport pavements refer to the *MTD (ETD)* parameter.

**Table 5.** Texture depth requirements for airfield pavements in operation.

| Aviation Organization | Runway Classification | Texture Depth (mm) |
|---|---|---|
| ICAO, EASA | A | 0.10–0.14 |
|  | B | 0.15–0.24 |
|  | C | 0.25–0.50 |
|  | D | 0.51–1.00 |
|  | E | 1.01–2.54 |

Measurements of the coefficient of friction and surface texture have been of high importance for the past 60 years or so. During this period, many different types of devices have been developed and used to measure these properties, which differ in terms of measurement principles, procedures and the way data is processed and reported. Research on the anti-skid properties of airport and road pavements have been and still are carried out by scientific centres, state aviation and road institutions around the world. As a result, there are numerous documents that include a review of measurement methods for anti-skid properties, as well as the results of research carried out in this area.

In 1992, the Permanent International Association of Road Congresses (PIARC) initiated a harmonization test program that consisted of a series of field tests to compare pavement texture and roughness [10]. The overall goal was to compare and harmonize the measurements of texture and surface friction coefficient. On the basis of the tests, a number of relations were developed including a common, harmonized index for wet surfaces, which was called the International Friction Index (IFI). Returning, however, to the measurement methods in the field of pavement texture, it could be measured in many ways, including, inter alia, using the volumetric technique and the water discharge rate technique [11]. Texture measurement devices (requiring lane closure) include the sand patch method (SPM) according to ASTM E965-15 Standard Test Method for Measuring Pavement Macro-Texture Depth Using a Volumetric Technique [12], OFM (outflow meter) according to ASTM E2380/E2380M-15 Standard Test Method for Measuring Pavement Texture Drainage Using an Outflow Meter [13] and circular texture meter CTM (circular texture meter) according to ASTM E2157-15 Standard Test Method for Measuring Pavement Macrotexture Properties Using the Circular Track Meter [14]. Rapid surface texture characterization methods are typically based on non-contact surface profiling techniques. An example of a non-contact profiler for describing pavement surface texture is the road surface analyzer (ROSANV), developed by the Federal Highway Administration (FHWA). ROSANV is a portable, automated system for measuring pavement texture at auto-straddle speeds along a linear path. ROSANV is equipped with a laser sensor mounted on the front bumper of the vehicle, and the device can operate at a speed of up to 113 km/h. The system calculates both the *MPD* and the estimated mean texture depth *MTD*, which is an estimate of the *MTD* obtained from the *MPD* using the transformation equation (1). As can be seen, many studies have been carried out to correlate different friction and texture measurement techniques, and the established correlations are important in order to determine the effects of microtexture and macrotexture on pavement and tyre friction under different pavement conditions. EASA, on the other hand, commissioned an external organization to conduct a runway friction test, cycle and braking tests. The overall aim of the study was to provide recommendations for the assessment of runway friction properties and runway condition reporting (RCR), the result of the project was the creation of a four-volume series of reports, i.e.,: Volume 1—Summary of Findings and Recommendations [15]; Volume 2—Documentation and Taxonomy [16]; Volume 3—Functional Friction [17] and Volume 4—Operational Friction [18].

When analyzing the available scientific publications on anti-skid properties research, it should be stated that this is a research area that attracts the attention of many research centres, mainly universities. Importantly, the essence of anti-skid properties in terms of air or road transport safety is noticed all over the world. In the work [19], the author points out that from the point of view of traffic safety, already at the stage of designing the

composition of mineral-bitumen mixtures for the wearing course, certain criteria should be taken into account, which would guarantee the maintenance of the required level of anti-skid properties. In turn, the work [20] presents selected devices for the assessment of anti-skid properties of road surfaces used in Poland and other countries. As shown, the DFT and CTM devices constitute a measurement set that allows for the independent assessment of macrotexture and microtexture and at the same time enables the assessment of anti-skid properties of road surfaces. In another work [21], on the basis of the conducted field research, the established functional relationships between the friction coefficient $\mu$, and the DFT20 coefficient and the *MPD* parameter were presented. The test results also indicated the possibility of using the mobile TWO measurement set and stationary CTM and DFT devices to assess the anti-skid properties of road surfaces. On the other hand, in [22], the authors presented the results of the research carried out with the use of CTM and DFT devices, but the SRT-3 device was used instead of the TWO device. They found that the SRT-3 device did not allow for a comprehensive evaluation of the surface in terms of skid resistance, therefore macrotexture evaluation by high-speed devices such as CTM and DFT should be performed. In turn, in [23] the author presents a comparison of changes in the anti-skid properties of asphalt concrete pavements (wearing layers and asphalt concrete) in the initial period of their operation, depending on the traffic load and the place of their installation. The analysis of anti-skid properties was carried out on the basis of BPN and macrotexture measurements by the volumetric method (point measurements), which showed that the intensity of the phenomena occurring on the pavement in the initial period after its construction significantly affects the friction coefficient and macrotexture. A similar theme was raised by a wider team of authors in the work [24]. The authors presented an assessment of the anti-skid properties in the initial period of pavement use on the basis of macrotexture and microtexture changes determined with the use of CTM and DFT devices (point measurements). On the basis of the obtained test results, they found that the most significant changes in the value of the friction coefficient take place immediately after the pavement is put to use. It has been shown that in the case of concrete pavements with exposed aggregate, changes in the anti-skid properties in the initial period of pavement use are related only to the microtexture of coarse aggregate grains. Based on measurements with the same equipment, but in the area of pavements made in the asphalt concrete technology, the authors presented the conclusions of the research in [25]. In [26], the authors proposed a new algorithm to evaluate a more reliable 3D macrotexture index assessed directly from a 2D profile due to the fact that the comparison of macrotexture values estimated on the basis of various measurement techniques usually provides poor agreement and unsatisfactory confidence in the actual macrotexture estimates using medium texture depth (*MTD*). Other authors, in [27] presented the concept of non-contact skid resistance measurement, which is based on the measurement of optical texture and consists of two elements: surface texture measurement using an optical measuring system and calculation of skid resistance based on the measured texture using a rubber friction model. Using new technical solutions, in [28] the authors presented an experiment confirming that the wrong choice of a sensor (for scanning 3D lines) with a long exposure time will significantly affect the measured texture data. In order to meet the global trends in the use of new measurement techniques in the field of texture, in [29] the authors raised the topic of the existing advantages of measurements performed with the use of 3D laser scanning technology. Some also attempted to find a correlation between texture and pavement friction [30]. Similar attempts in this regard were presented in [31], where the 3D laser imaging technology was used.

When reviewing the literature in the area in question, it can be seen that the work is primarily focused on obtaining the possibility of predicting pavement roughness (friction coefficient) based on independent texture tests (point measurements) conducted in laboratory and field conditions. Research centres focus on conducting research in this area, more than once as part of assigned research projects. At the same time, it should be noted that most of the research work carried out so far is based on measurements carried out on

road surfaces, not airport surfaces, and some of them are even limited to measurements in laboratory conditions. On the other hand, the existing reference documents concerning the texture of airport surfaces, in practice, raise many doubts and they concern mainly the runway. Moreover, these documents do not specify the minimum values, as is the case with the coefficient of friction for particular age ranges of the pavement, i.e., "values for new pavements", "values for pavements in operation, beyond which corrective actions should be taken" and "minimum limit values" (in other words: new pavements "N"/pavements in operation "E"/surfaces intended for renovation "S"). In view of the above, the authors analyzed the obtained results of texture tests for new airport pavements in relation to the current requirements. An innovative method of assessing anti-skid properties was used for the tests, in which a new coefficient of continuous mean profile depth and texture *CMPTD* is determined, the value of which can be converted into the mean profile depth, *MPD,* and the estimated texture depth, *ETD*. The main aim of the work is to discuss and compare the currently used texture requirement for new airport pavements (1.00 mm) with the requirement proposed by the authors, determined on the basis of actual results from field measurements.

## 2. Materials and Methods

The tests were carried out with the use of a measuring system built on the basis of the ASFT airport surface friction tester (ASFT, Ystad, Sweden) on a T-10 trailer, additionally equipped with a 2D/3D high-frequency laser scanner (Micro Epsilon Messtechnik, Ortenburg, Germany)—measuring in the traces of the friction tester's measuring wheel). Similar attempts are also made by others, not mentioned earlier [32–35], but the research methods adopted by them do not include the simultaneous measurement of these two parameters. It should also be noted that, in contrast to the currently used measurement method (profilometric), the proposed method of measuring the texture depth as part of the evaluation of the anti-skid properties of airport pavements allows for measurements both in the direction parallel and perpendicular to the direction of the friction tester. This method is not limited by the spot measurement, i.e., the measurement is not carried out only as a function of the length of the measurement section, but also as a function of its width (corresponding to the width of the contact between the tyre of the friction tester and the road surface). The measuring system for the adopted method of assessing the anti-skid properties includes (Figure 1):

- Roughness evaluation module—measurement of the friction coefficient;
- Texture depth assessment module—measurement of the new *CMPTD* coefficient defining the continuous average depth of the profile and texture.

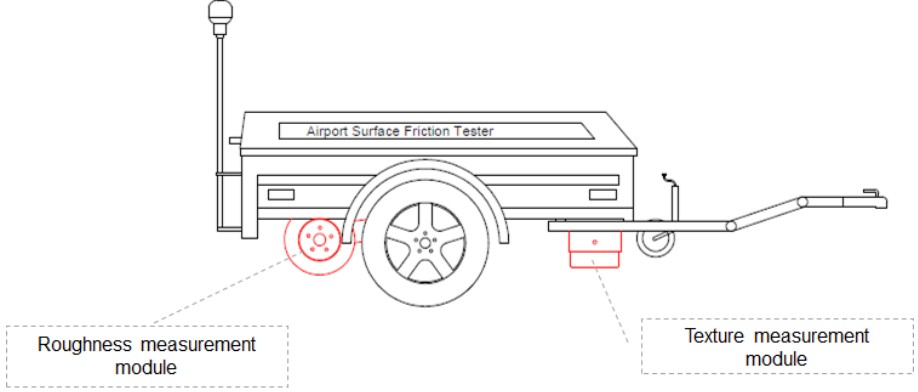

**Figure 1.** Scheme of measurement system for anti-skid properties assessment.

Roughness evaluation module is the airfield pavement friction tester on trailer T-10 manufactured by ASFT (Ystad, Sweden) (Figure 2).

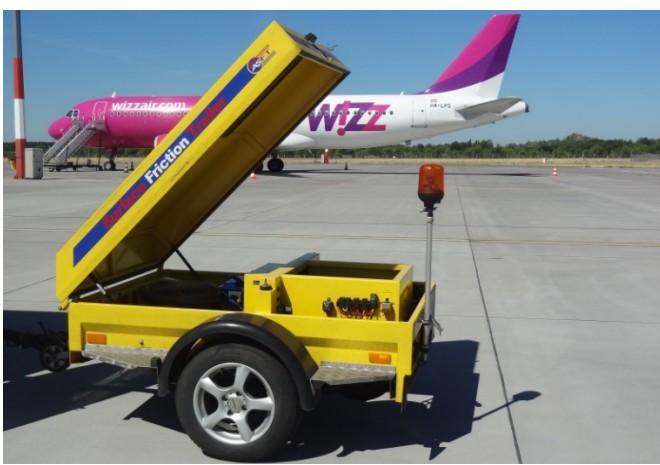

**Figure 2.** Airport Surface Friction Tester on Trailer T-10.

The texture depth assessment module is a 2D/3D profile scanner of the scan-CONTROL HIGH-SPEED 2660-100 (LLT 2660-100) type by Micro-Epsilon (Ortenburg, Germany) (Figure 3), enabling non-contact measurement of geometric quantities [36].

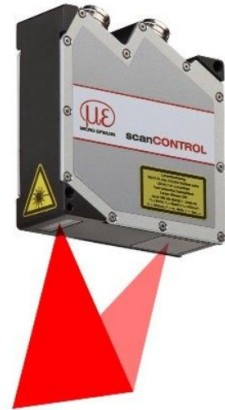

**Figure 3.** scanCONTROL profile scaner.

Test measurements were carried out at selected airport facilities, at measurement speeds of 65 and 95 km/h, on surfaces of all ages made both in the technology of cement concrete and asphalt concrete. The choice of airport (test) facilities was determined by the availability of specific surfaces (age, technology of execution). The pavements for each age range were adopted on the basis of their actual service life (age), i.e.,: for "N" up to 3 years, for "E" from 3 to 20 years and for "S" over 20 years. However, the work focuses only on the analysis of the results for new pavements. In accordance with the adopted field research plan, for each of the specified measurement conditions (e.g., measuring speed 65 km/h, new pavement, pavement made in the asphalt concrete technology), 6 measurement sections were determined, each 100 m long. Sections for specific measurement conditions were located on one AFE, for example, the taxiway. The individual sections were treated as a single sample, while the 6 sections for the given measurement conditions were treated as a population of samples. The measurement of the *CMPTD* and $\mu$ coefficients (built measuring system) was performed continuously along the entire length of the measuring section, with the reading frequency every 0.2 m for *CMPTD* and 10 m for $\mu$. Therefore, for further analyses, the results from 10 m sections for individual measurement sections were adopted. In addition, on each section, point measurements were also made with the ELATextur device (at 20, 40 and 60 m of the measuring section). The view of the performed measurements is shown in Figures 4 and 5, while Figure 6 presents the measurement diagram according to the adopted methodology.

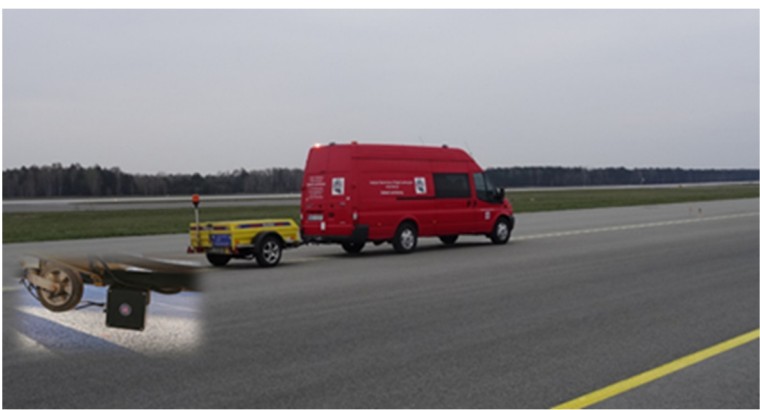

**Figure 4.** Test with the measurement module.

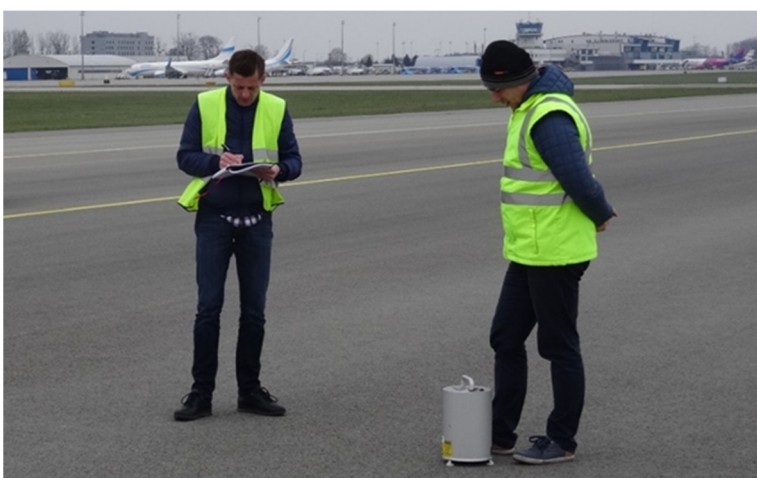

**Figure 5.** ELATextur measurement.

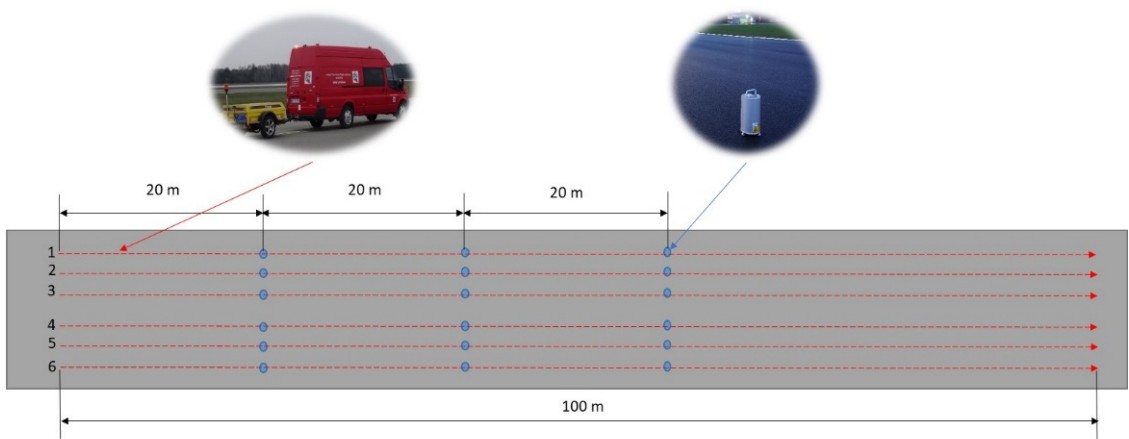

**Figure 6.** Scheme of the measurements in the field.

The analysis methodology of the test results included determining the value of the criterion for the *CMPTD* coefficient for new airport pavements (criterion independent of the measuring speed) and the determination of the transformation equations for the

transition from *CMPTD* to *MPD* and *ETD*. To determine the value of the criterion for the *CMPTD* coefficient, the values from field measurements of the $\mu$ and *CMPTD* coefficients and the known criteria for the $\mu$ coefficient (depending on the measuring speed) were used. On the other hand, to determine the transformation equations, the values from the field measurements of the *CMPTD*, *MPD* and *ETD* coefficients were used, between which the Pearson linear correlation coefficient was determined, which determines the level of the linear association.

Determination of the criterion for the *CMPTD* coefficient value was carried out using two methods, i.e.,:

- Two ratios put into an equation method—proportion (Figure 7);
- Value estimation method—predicting the y value at a given *x* value based on existing (known) *x* and *y* values (Figure 8).

In each method, two models were adopted, i.e.,: model I—analysis of the results for two speeds separately, model II—analysis of the results for two speeds together (65 and 95 km/h in one set).

In addition, for each model, two methods were adopted for assigning values from field measurements to individual age ranges of the pavement (in this case to the "N" range), i.e.,:

- "By age"—assignment of the (*CMPTD* and $\mu$) values obtained from the field tests on the basis of actual pavement age;
- "By value"—assignment of the (*CMPTD* and $\mu$) values obtained from the field tests on the basis of $\mu$ values, fulfilling required criteria for the specific age "N".

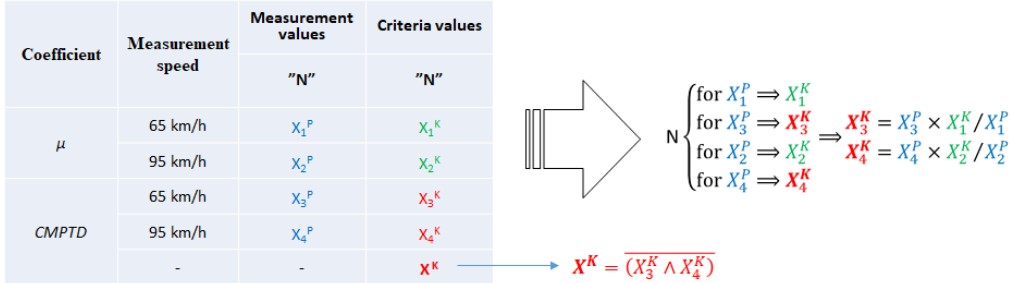

**Figure 7.** Determination of the search criteria—two ratios put into an equation method.

where:

$X_1^P$, $X_2^P$, $X_3^P$, $X_4^P$—mean values obtained during field measurements;

$X_1^K$, $X_2^K$—known criteria values for the $\mu$ coefficient;

$X_3^K$, $X_4^K$—the search criteria values for the *CMPTD* coefficient;

$X^K$—final, searched criterion value for the *CMPTD* coefficient.

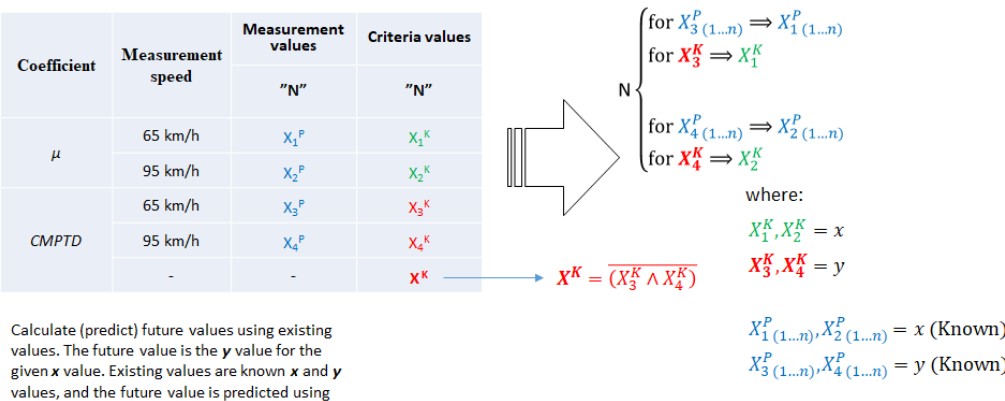

**Figure 8.** Determination of the search criteria—value estimation method.

where:

$X_1{}^P{}_{(1...n)}$—the set of individual values for the $X_1{}^P$ mean value;

$X_2{}^P{}_{(1...n)}$—the set of individual values for the $X_2{}^P$ mean value;

$X_3{}^P{}_{(1...n)}$—the set of individual values for the $X_3{}^P$ mean value;

$X_4{}^P{}_{(1...n)}$—the set of individual values for the $X_4{}^P$ mean value.

In addition, when determining the value of the criterion, weights were introduced, determining the validity of the adopted method of analyzing the results in terms of assigning values from field measurements to the age range of new pavements, i.e.,:

- "By age" method—0.20;
- "By value" method—0.80.

The weights were selected using the expert method. The basis for estimating the value of weights were the results of many years of diagnostic research in the field of measurement of the friction coefficient and pavement texture performed at civil and military airport facilities located in Poland.

## 3. Results

As a result of the field measurements performed, the values for the coefficients determining the anti-skid properties of airport pavements were measured according to the conditions specified for the measurements of the friction coefficient, i.e., for three age groups. As mentioned before, the analysis of the test results was limited only to new airport pavements. The obtained results are presented in the further part of the work, taking into account the adopted methodology of their analysis.

### 3.1. Criterion for the CMPTD Coefficient

The results of the field measurements of the $\mu$ and *CMPTD* coefficients used to determine the criterion value for the *CMPTD* coefficient are presented below in Tables 6–9.

**Table 6.** Field tests results for asphalt concrete—model I (two speeds separately).

| Coefficient | Test Speed (km/h) | Identification | Measurement Values (mm) | |
|---|---|---|---|---|
| | | | "By Age" | "By Value" |
| $\mu$ | 65 | $X_1{}^P$ | 0.68 | 0.76 |
| | 95 | $X_2{}^P$ | 0.80 | 0.74 |
| *CMPTD* | 65 | $X_3{}^P$ | 1.08 | 0.66 |
| | 95 | $X_4{}^P$ | 0.27 | 0.50 |

**Table 7.** Field tests results for asphalt concrete—model II (two speeds together—65 and 95 km/h in one collection).

| Coefficient | Test Speed (km/h) | Identification | Measurement Values (mm) | |
|---|---|---|---|---|
| | | | "By Age" | "By Value" |
| $\mu$ | 65 | $X_1{}^P$ | 0.74 | 0.74 |
| | 95 | $X_2{}^P$ | | |
| *CMPTD* | 65 | $X_3{}^P$ | 0.67 | 0.63 |
| | 95 | $X_4{}^P$ | | |

**Table 8.** Field tests results for cement concrete—model I (two speeds separately).

| Coefficient | Test Speed (km/h) | Identification | Measurement Values (mm) | |
|---|---|---|---|---|
| | | | "By Age" | "By Value" |
| $\mu$ | 65 | $X_1{}^P$ | 0.77 | 0.76 |
| | 95 | $X_2{}^P$ | 0.75 | 0.71 |
| *CMPTD* | 65 | $X_3{}^P$ | 0.28 | 0.30 |
| | 95 | $X_4{}^P$ | 0.10 | 0.28 |

**Table 9.** Field tests results for cement concrete—model II (two speeds together—65 and 95 km/h in one collection).

| Coefficient | Test Speed (km/h) | Identification | Measurement Values (mm) | |
|---|---|---|---|---|
| | | | "By Age" | "By Value" |
| $\mu$ | 65 | $X_1{}^P$ | 0.76 | 0.73 |
| | 95 | $X_2{}^P$ | | |
| *CMPTD* | 65 | $X_3{}^P$ | 0.19 | 0.29 |
| | 95 | $X_4{}^P$ | | |

However, Tables 10 and 11 show the determined values of the searched criterion for the *CMPTD* coefficient.

**Table 10.** Search criterion values for asphalt concrete.

| Analysis Method | Analysis Model | Identification | Search Criterion Values (mm) | | | |
|---|---|---|---|---|---|---|
| | | | "By Age" | "By Value" | Mean | |
| - | - | - | | | No Weights | With Weights |
| Proportion | Model I | | 0.66 | 0.50 | 0.58 | 0.54 |
| | Model II | | 0.59 | 0.56 | 0.58 | 0.57 |
| Estimation | Model I | $X^K$ | 0.70 | 0.68 | 0.69 | 0.68 |
| | Model II | | 1.14 | 0.83 | 0.99 | 0.89 |
| Mean: | | | 0.77 | 0.64 | 0.71 | 0.67 |

**Table 11.** Search criterion values for cement concrete.

| Analysis Method | Analysis Model | Identification | Search Criterion Values (mm) | | | |
|---|---|---|---|---|---|---|
| | | | "By Age" | "By Value" | Mean | |
| | | | | | No Weights | With Weights |
| - | - | - | | | | |
| Proportion | Model I | | 0.17 | 0.26 | 0.21 | 0.24 |
| | Model II | | 0.16 | 0.26 | 0.21 | 0.24 |
| Estimation | Model I | $X^K$ | 0.19 | 0.36 | 0.27 | 0.32 |
| | Model II | | 0.14 | 0.34 | 0.24 | 0.30 |
| Mean: | | | 0.16 | 0.30 | 0.23 | 0.28 |

The performed field tests and then the analysis of the obtained results according to the adopted methodology clearly indicate that when assessing the airport pavement in terms of its texture, one should, first of all, distinguish between whether the assessment concerns the surface made in the cement concrete technology or asphalt concrete. At the same time, based on the results presented above, the authors would like to propose a criterion for a new *CMPTD* coefficient in relation to new airport surfaces, i.e.,:

- For asphalt concrete pavements—0.67 mm;
- For pavements made in the technology of cement concrete—0.28 mm.

*3.2. Transformation Equations*

The results of the field measurements of the *CMPTD*, *MPD* and *ETD* coefficients for the transformation equations determination are presented below in Tables 12–15.

**Table 12.** Field tests results for asphalt concrete (65 km/h).

| Test Speed (km/h) | Pavement Construction Technology | Strip (Sample) | Measurement Values (mm) | | |
|---|---|---|---|---|---|
| | | | *CMPTD* | *MPD* | *ETD* |
| 65 | Asphalt concrete | 1 | 1.05 | 0.81 | 0.85 |
| | | 2 | 1.13 | 0.93 | 0.94 |
| | | 3 | 1.03 | 1.11 | 1.09 |
| | | 4 | 1.21 | 1.30 | 1.28 |
| | | 5 | 0.95 | 1.04 | 1.03 |
| | | 6 | 1.11 | 0.89 | 0.91 |
| | Mean: | | 1.08 | 1.01 | 1.02 |

**Table 13.** Field tests results for asphalt concrete (95 km/h).

| Test Speed (km/h) | Pavement Construction Technology | Strip (Sample) | Measurement Values (mm) | | |
|---|---|---|---|---|---|
| | | | *CMPTD* | *MPD* | *ETD* |
| 95 | Asphalt concrete | 1 | 0.27 | 0.72 | 0.75 |
| | | 2 | 0.26 | 0.66 | 0.72 |
| | | 3 | 0.26 | 0.77 | 0.76 |
| | | 4 | 0.32 | 0.76 | 0.80 |
| | | 5 | 0.25 | 0.73 | 0.78 |
| | | 6 | 0.28 | 0.62 | 0.65 |
| | Mean: | | 0.27 | 0.71 | 0.74 |

**Table 14.** Field tests results for cement concrete (65 km/h).

| Test Speed (km/h) | Pavement Construction Technology | Strip (Sample) | Measurement Values (mm) | | |
|---|---|---|---|---|---|
| | | | *CMPTD* | *MPD* | *ETD* |
| 65 | Cement concrete | 1 | 0.27 | 0.42 | 0.54 |
| | | 2 | 0.26 | 0.45 | 0.56 |
| | | 3 | 0.28 | 0.59 | 0.67 |
| | | 4 | 0.28 | 0.55 | 0.64 |
| | | 5 | 0.29 | 0.77 | 0.82 |
| | | 6 | 0.28 | 0.37 | 0.50 |
| | Mean: | | 0.28 | 0.53 | 0.62 |

**Table 15.** Field tests results for cement concrete (95 km/h).

| Test Speed (km/h) | Pavement Construction Technology | Strip (Sample) | Measurement Values (mm) | | |
|---|---|---|---|---|---|
| | | | *CMPTD* | *MPD* | *ETD* |
| 95 | Cement concrete | 1 | 0.08 | 0.28 | 0.43 |
| | | 2 | 0.10 | 0.26 | 0.41 |
| | | 3 | 0.10 | 0.25 | 0.40 |
| | | 4 | 0.06 | 0.24 | 0.39 |
| | | 5 | 0.11 | 0.23 | 0.39 |
| | | 6 | 0.15 | 0.24 | 0.39 |
| | Mean: | | 0.10 | 0.25 | 0.40 |

Figures 9 and 10 below show the histograms with the linear correlation coefficients determining the level of the linear association between the *CMPTD* and *MPD* and *ETD* coefficients (confidence interval 0.95).

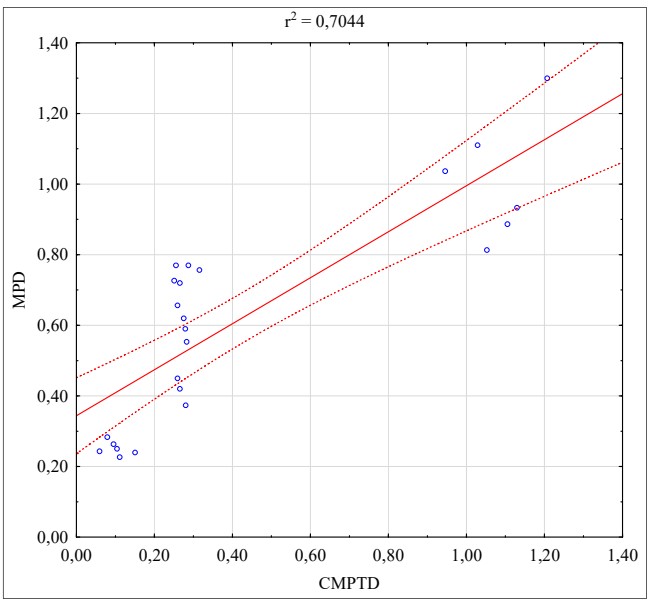

**Figure 9.** The level of linear association between the *CMPTD* and *MPD*.

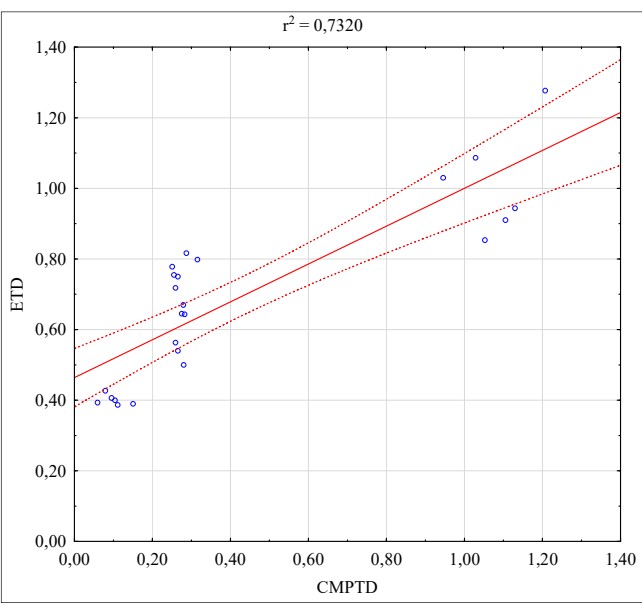

**Figure 10.** The level of linear association between the *CMPTD* and *ETD*.

However, the transformation equations determined on this basis for the transition from *CMPTD* to *MPD* and *ETD* are as follows:

$$MPD = 0.65 \times CMPTD + 0.34 \tag{2}$$

$$ETD = 0.54 \times CMPTD + 0.46 \tag{3}$$

Due to the determined transformation equations, after performing the measurements with the use of the proposed method of testing the anti-skid properties, the *MPD* and *EDT* coefficients can be determined on the basis of which the airport pavement can be assessed in relation to the current requirements.

Nevertheless, the derived equations also offer the possibility of proposing a new mean texture depth (*ETD*) criterion for new airport surfaces. i.e.,:

- For asphalt concrete pavements—0.82 mm;
- For cement concrete pavements—0.61 mm.

## 4. Discussion

From a practical point of view, the topic raised by the authors is extremely important, primarily in terms of the safety of air operations. The aim is all the time to ensure that airfield pavements have good anti-skid properties, which should be defined by specific (required) values, in this case, the friction coefficient and the texture depth.

The conducted field tests and then the analysis of the obtained test results according to the adopted methodology clearly show that obtaining the average texture depth value required for new airfield pavements (currently 1.00 mm) is unlikely in practice (Tables 12–15). The obtained results clearly indicate that when assessing the airfield pavement in terms of its texture, one should, first of all, distinguish whether the assessment concerns the cement concrete or asphalt concrete surface. As can be seen, both in the case of the pavement assessment with the measuring method (*MPD* and *ETD*), as well as the proposed new measurement method (*CMPTD*), the values of the coefficients characterising the asphalt concrete pavement texture are correspondingly high er, whereas in the cement concrete pavements the values are correspondingly lower. According to the authors, it would be necessary to consider the introduction of separate requirements in this respect with regard to the airfield pavement construction technology. Therefore, the authors allowed themselves to propose new medium texture depth (*ETD*) criteria for new airport surfaces, i.e.:

- For pavements made in asphalt concrete technology—0.82 mm;
- For pavements made in cement concrete technology—0.61 mm.

Maybe it is also worth introducing a new CMPTD coefficient for continuous mean depth profile and texture measurement, which makes it possible to obtain information on the texture/profile of the pavement in a short time and, above all, in the entire length of the tested AFL? The proposed new measurement method, compared to the currently used (point) methods, gives greater possibilities. From the research point of view, it enables, first of all, the continuous measurement of the pavement texture, which, as can be seen, is not without significance when comparing the values from field measurements (visible difference in values). Taking into account the results obtained so far and presented, the authors plan further work related to the analysis of the results in relation to the remaining age ranges of the pavement. The results of research and analyses in this area will supplement the current set of works [37–40] and will constitute the basis for further studies related to the subject of airfield pavement texture.

**Author Contributions:** Conceptualization, M.Z. and M.W.; methodology, M.W. and K.B.; formal analysis, resources, data curation, writing—original draft preparation, K.B.; writing—review and editing, M.W. and P.I. All authors have read and agreed to the published version of the manuscript.

**Funding:** The research was funded by the Air Force Institute of Technology as part of the allocated funding from the own research fund for the research task "Construction of a measuring system for testing the texture of airfield pavements in terms of skid resistance properties and their impact on the safety of air operations".

**Institutional Review Board Statement:** Not applicable.

**Informed Consent Statement:** Not applicable.

**Data Availability Statement:** Data sharing not applicable.

**Conflicts of Interest:** The authors declare no conflict of interest.

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
