# Peer review of "Analysis of the Anti-Skid Properties of New Airfield Pavements in Aspect of Applicable Requirements"

_coatings, doi:10.3390/coatings11070778_

Round 1
Reviewer 1 Report
Regarding the methodology, it is unclear that how the test facilities and test points were chosen (second paragraph of page 8). More details about the sampling or sample selection should be given.
What are the limitations of the research? How would these limitations affect the interpretation of the research findings?
The use of language in the paper should be improved. The paper has quite a number of grammatical errors. The paper should have been proofread by a professional English writer before submission.
The authors should make sure the short-forms used have been well explained. Short-forms like "ASFT" and "T2Go" have not been explained.
The structure of the paper is weird. I would suggest the authors to avoid using short paragraphs. These short paragraphs should be linked or combined.
Presentation of the tables can be improved as well. For example in the first column of Table 1, it is better to have "0.36 to 0.39" rather than the other way round. Also, it should be "0.40 or more" and "0.25 or less" instead of "0.40 and more" and "0.25 and less".
Author Response
- The selection of airport (test) and measuring sections (samples) has been detailed.
- In this study, the only limitation but not affecting the interpretation of the results is the need to obtain the required measuring speed (65 km/h or 95 km/h). Therefore, the measurement method presented cannot be used on aerodrome functional elements whose geometric dimensions do not allow the measuring system to achieve the required speed. In such cases, point measurement methods shall be used.
- The article has been validated for language validation.
- The abbreviation ASFT has been clarified, while T2Go is the name of the measuring device.
- The number of short paragraphs has been reduced.
- Fixed the description in Table 1 by source document.
Reviewer 2 Report
This manuscript does not correspond to the high-quality criteria of the journal of Coatings. The scope of this manuscript is different from the journal's scope. Therefore, I recommend rejecting this manuscript.
This is clearly a student’s paper. Although it is adequately written, it offers no critical information and no new slant on the review topic. Most of the content in the manuscript is better for the journal with a lower level.
Author Response
- According to the authors, the article is part of the issue of the special magazine Coatings "Road Surface Performance: Skid Resistance, Noise, and Rolling Resistance". The article presents an innovative method for assessing the anti-skid properties of airfield pavements. The analysis of the results of field studies carried out by the authors and presented in the article made it possible to draw conclusions relevant from the practical point of view of the operation of airfield pavements. The authors have been practically engaged in the broader topic of airfield pavement diagnostics for several years.
Reviewer 3 Report
The scanning device is useful for inspection of actual airfield runways, but I feel the academic originality/findings are weak.
My comments and suggestions are shown below, even with my weak understanding.
Page 1, abstract:
Are the words "MPD and ETD" well known? Un-known words should not be used in the abstract.
--------------------------------------------
Page 1, abstract:
What was the doubt indicated from the tests? Detailed explanation is needed in the abstract.
--------------------------------------------
Page 7, Fig.1
I am not sure how the module measures roughness of pavements, because of my weak understanding. Can it measure roughness even below 10 m intervals at a speed of 95 km/h?
--------------------------------------------
Page 8, line 281:
"However, the work focuses only on the analysis of the results for new pavements."
The reason should be described simply. In addition, comments about applicability for old pavements are also needed in the later section (e.g. conclusion section).
--------------------------------------------
Page 10, lines 334-337
“The basis for estimating the value of weights -- in Poland”.
Can the database and weight values apply to airfield runways in other countries? In other words, materials and surface properties of the pavements are common in the world?
--------------------------------------------
Page 10, Figs. 7 and 8
Additional/detailed explanations for Figs. 7 and 8 are needed in Section 2, to understand the figures and methods easily.
--------------------------------------------
Page 14, line 396:
Section 4 “Discussion”
What are the conclusions and findings observed from the study? It is recommended to create a new conclusion section, instead of Section 4 "Discussions". Some descriptions and suggestions are already shown in Section 3, so I think Section 4 is better to be compressed and changed to the conclusion section.
--------------------------------------------
Page 14, line 396:
Section 4 “Discussion”
The testing device is useful for inspection of airfield runways. On the other hand, I could not find new findings from an academic view point? Could you describe/appear the academic originality in the conclusions?
Author Response
- MPD and ETD shortcuts are explained.
- The research (tests) do not raise any doubts. While the obtained results (both the proposed method and the method currently used point) show that the acquisition required for the new airport pavement values of average texture depth (currently 1.00 mm) is unlikely. According to the authors indicate doubt and its explanation in the summary is sufficient.
- The measurement is carried out continuously, regardless of the measurement speed, while the measuring device (in this case the ASFT friction tester on a T-10 trailer) generates results with a frequency of 10 m.
- Exactly, the work focuses solely on the analysis of the results for new airport pavements and this is the intended approach of the authors. At the same time, the paper indicates the direction of further work in this area, i.e. the analysis of the results for pavements in the remaining age ranges.
- The values proposed in the article refer to airport pavements, regardless of the country of their occurrence.
- Explanations under the figures have been extended.
- Section 4 is mandatory. The authors do not see the need to create a "summary" section as section 4 is not very extensive. Conclusions from the conducted research are included in section 3, as well as in section 4.
- The article presents an innovative method of assessing the anti-skid properties of airfield pavements. The novelty of this method consists in the simultaneous and dynamic measurement of the parameters determining the anti-skid properties of airfield pavements, i.e. the friction coefficient and the depth of texture / pavement profile. This method uses a proprietary measuring system. So far in the world no such attempts have been made to simultaneously and exactly in the same trace (tire trace of the friction tester measuring wheel) to measure these two parameters. According to the authors, the content of the article fully reflects its originality.
Round 2
Reviewer 1 Report
I think the authors have addressed my concerns to my satisfaction. I don't have any further comments on the paper.
Reviewer 2 Report
This manuscript does not correspond to the high-quality criteria of the journal of Coatings. The scope of this manuscript is different from the journal's scope. Therefore, I recommend rejecting this manuscript.
This is clearly a student’s paper. Although it is adequately written, it offers no critical information and no new slant on the review topic. Most of the content in the manuscript is better for the journal with a lower level.
Reviewer 3 Report
I have checked the revised paper.